# FLASHBACK: UNDERSTANDING AND MITIGATING FORGETTING IN FEDERATED LEARNING

## ABSTRACT

In Federated Learning (FL), forgetting, or the loss of knowledge across rounds, hampers algorithm convergence, especially in the presence of severe data heterogeneity among clients. This study explores the nuances of this issue, emphasizing the critical role of forgetting leading to FL's inefficient learning within heterogeneous data contexts. Knowledge loss occurs in both client-local updates and server-side aggregation steps; addressing one without the other fails to mitigate forgetting. We introduce a metric to measure forgetting granularly, ensuring distinct recognition amid new knowledge acquisition. Based on this, we propose Flashback, a novel FL algorithm with a dynamic distillation approach that regularizes the local models and effectively aggregates their knowledge. The results from extensive experimentation across different benchmarks show that Flashback mitigates forgetting and outperforms other state-of-the-art methods, achieving faster round-to-target accuracy by converging in 6 to 16 rounds, being up to $27\times$ faster.

## 1 INTRODUCTION

Federated Learning (FL) is a distributed learning paradigm that allows training over decentralized private data. These datasets belong to different clients that participate in training a global model. Federated Averaging (FedAvg) (McMahan et al., 2017) is a prominent training algorithm that uses a centralized server to orchestrate the process. At every round, the server samples a proportion of the available clients. Starting from the current version of the global model, each sampled client performs $E$ epochs of local training using their private data and sends its updated model to the server. Then, the server aggregates the models by averaging them to obtain the new global model. This process is typically repeated for many rounds until a desired model performance is obtained.

A main challenge in FL is the data heterogeneity in distribution between the clients' private datasets, which are unbalanced and non-IID (Kairouz et al., 2021). Data heterogeneity causes local model updates to drift – the local optima might not be consistent with the global optima – and can lead to slow convergence of the global model – where more rounds of communication and local computation are needed – or worse, the desired performance may not be reached. Addressing data heterogeneity in FL has been the focus of several prior studies. For instance, FedProx (Li et al., 2020) proposes a proximal term to limit the distance between the global model and the local model updates, mitigating the drift in the local updates. MOON (Li et al., 2021b) mitigates the local drift using a contrastive loss to minimize the distance between the feature representation of the global model and the local model updates while maximizing the distance between the current model updates and the previous model updates. FedDF (Lin et al., 2020) addresses heterogeneity in local models by using ensemble distillation during the aggregation step (instead of averaging the model updates). Nonetheless, we experimentally observe that under severe data heterogeneity, these proposals provide little or even no advantage over FedAvg. Figure 1a illustrates the test accuracy of FedAvg and other baselines while training a DNN over the CIFAR10 dataset (Krizhevsky, 2009) (more details are in § 5).

This motivates us to understand better how data heterogeneity poses a challenge for FL and devise a new approach to handling non-IID datasets. We investigate the evolution of the global model accuracy broken down by its per-class accuracy. Figure 1b shows a heatmap of the per-class accuracy for FedAvg; each rectangle represents the accuracy of the global model on a class at a round. Other baseline methods show similar results. Our key observation is that there is a significant presence of *forgetting*: i.e., cases where some knowledge obtained by the global model at round $t$ is forgotten

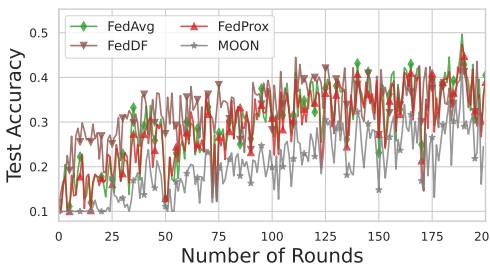

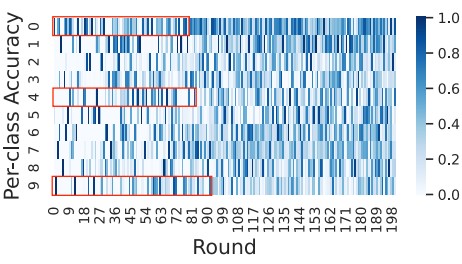

(a) Global model accuracy of FedAvg and baselines.

(b) Per-class accuracy of FedAvg's global model.

Figure 1: Performance of FedAvg and other methods over training rounds with CIFAR10.

at round $t + 1$, causing the accuracy to decline (as shown by the prominent number of light-shaded rectangles at the right side of darker ones in the figure; we highlight some cases in red in Fig. 1b).

A similar phenomenon is known as *catastrophic forgetting* in Continual Learning (CL) literature (Parisi et al., 2019). CL addresses the challenge of sequentially training a model on a series of tasks, denoted as $\{T_1, T_2, \ldots, T_n\}$, without revisiting data from prior tasks. Formally, given a model with parameters $\theta$ and task-specific loss functions $L_t(\theta)$ for each task $T_t$, the objective in CL is to update $\theta$ such that performance on the current task is optimized without significantly degrading the model's performance on previously learned tasks. This is non-trivial, as naïve sequential training often leads to catastrophic forgetting, where knowledge from prior tasks is overridden when learning a new task. An inherent assumption in this paradigm is that once the model transitions from task $T_i$ to task $T_{i+1}$, data from $T_i$ becomes inaccessible, amplifying the importance of knowledge retention strategies (De Lange et al., 2021).

While the premises and assumptions of FL differ from those of traditional machine learning and continual learning, forgetting remains an issue. This can be viewed as a side effect of data heterogeneity, a commonality FL shares with CL. In FL, the global model evolves based on a fluctuating data distribution. Specifically, a diverse set of sampled clients with distinct data distributions contribute a model update in each communication round. Furthermore, these model updates must be aggregated to obtain a global model. This situation presents dual-levels of data heterogeneity. Firstly, at the *intra-round* level, heterogeneity arises from the participation of clients with varied data distributions within the same round. This diversity can inadvertently lead to "forgetting" specific data patterns or insights from certain clients. Secondly, at the *inter-round* level, the participating clients generally change from one round to the next. As a result, the global model may "forget" or dilute insights gained from clients in previous rounds.

To remedy this issue, we propose Flashback, a FL algorithm that employs a dynamic distillation approach to mitigate the effects of data heterogeneity. Flashback's dynamic distillation ensures that the local models learn new knowledge while retaining knowledge from the global model during the client updates by adaptively adjusting the distillation loss. Moreover, during the server update, Flashback uses a very small public dataset as a medium to integrate the knowledge from the local models to the global model using the same dynamic distillation. Flashback performs these adaptations by estimating the knowledge in each model using label counts as a proxy of the model knowledge. Overall, Flashback results in a more stable and faster convergence compared to existing methods.

Our contributions are the following:

- We systematically investigate the forgetting problem in FL. We show that under severe data heterogeneity, FL sufferers from forgetting. We dissect how and where forgetting happens (§ 3).

- We propose a new metric for measuring forgetting over the communication rounds (§ 3).

- We introduce *Flashback*, a FL algorithm that employs a dynamic distillation during the local updates and the server update (§ 4). By addressing the forgetting issue, Flashback mitigates its detrimental effects and converges to the desired accuracy faster than existing methods (§ 5)

## 2 BACKGROUND

We consider a standard cross-device FL setup in which there are $N$ clients. Each client $i$ has a unique dataset $D_i = \{(x_j, y_j)\}_{j=1}^{n_i}$ where $x_j$ represents the input features and $y_j$ is the ground-truth label for $j$-th data point and $n_i$ represent the size of the local dataset of client $i$. The goal is to train a single global model that minimizes the objective:

$$\min_{w \in \mathbb{R}^d} \sum_{i=1}^{N} \frac{|D_i|}{|\cup_{i \in [N]} D_i|} \left\{ L_i(w) = \frac{1}{|D_i|} \sum_{j=1}^{|D_i|} l(w; (x_j, y_j)) \right\},$$

where $L_i(w)$ represents the local loss for client $i$, and $l(w; (x_j, y_j)) = \mathcal{L}_{\text{CE}}(F_w(x), y)$ is the cross-entropy loss for a single data point, where $F_w$ denotes the model parameterized by learnable weights $w$.

FedAvg provides a structured approach to address this distributed problem efficiently. At each communication round $t$, the server randomly selects $K$ clients from the total available $N$ clients. These clients (denoted with $\mathbb{S}_t$) receive the previous global model, $w_{t-1}$. Then, they update this model based on their local data using their local loss function $L_i$. After updating, each client sends their modified model $w_{k,t}$ back to the server that updates the global model using a weighted average of local models, i.e., $w_t = \sum_{k \in \mathbb{S}_t} \frac{|D_k| w_{k,t}}{|\cup_{k \in [K]} D_k|}$. Various FL algorithms introduce modifications at the local update level or during the global aggregation to accommodate the intrinsic heterogeneity in client data. The nuances of these variations are further explored in § 6.

Among these, **Knowledge Distillation (KD)** is a training method wherein a smaller model, referred to as the student, is trained to reproduce the behavior of a more complex model or ensemble called the teacher. Let $F_{w_s}$ denote the student model with weights $w_s$ and $F_{w_t}$ represent the teacher model with weights $w_t$. For a given input $x$, the student aims to minimize the following distillation loss:

$$\begin{aligned} \mathcal{L}_{\text{KD}}((x, y); w_s, w_t) = &\mathcal{L}_{\text{CE}}(F_{w_s}(x), y)(1 - \alpha) \\ &+ \mathcal{L}_{\text{KL}}(F_{w_t}(x), F_{w_s}(x))\alpha \end{aligned} \tag{1}$$

Here, $\mathcal{L}_{\text{CE}}$ is the standard cross-entropy loss with true label $y$, and $\mathcal{L}_{\text{KL}}$ represents the Kullback-Leibler (KL) divergence between the teacher's and the student's output probabilities. It is defined as $\mathcal{L}_{\text{KL}}(\boldsymbol{p}, \boldsymbol{q}) = \sum_{c=1}^{C} p^c \log\left(\frac{p^c}{q^c}\right)$, where $C$ is the number of classes, $\mathbf{p}$ is the target output probability vector, and $\mathbf{q}$ is the predicted output probability vector. The hyperparameter $\alpha \in [0, 1]$ balances the importance between the learning from the true labels and the teacher's outputs.

While distillation originally emerged as a method for model compression Hinton et al. (2015); Buciluǎ et al. (2006); Schmidhuber (1991), its utility extends to FL. In the federated context, distillation can combat challenges like data heterogeneity Lin et al. (2020); Lee et al. (2021) and communication efficiency Jeong et al. (2018).

## 3 FORGETTING IN FL

We now investigate where forgetting happens and devise a metric to quantify this phenomenon. Recall that in FL, the models are updated in two distinct phases: 1) during local training – when each client $k$ starts from global model $w_{t-1}$ and locally trains $w_{k,t}$ – and 2) during the aggregation step – when the server combines the client models to update the new global model $w_t$.

Intuitively, forgetting in FL is when knowledge contained in the global model will be lost after the completion of communication round $w_{t-1} \to w_t$. We observe that forgetting may occur in the two phases of FL. We refer to the former case as **local forgetting**, where some knowledge in the global model will be lost during the local training $w_{t-1} \to w_{k,t}$. This is due to optimizing for the clients' local objectives, which depend on their datasets. Local forgetting is akin to the form of forgetting seen in CL, where tasks change over time (as with clients in FL) and, consequently, the data distribution. We refer to the latter case as **aggregation forgetting**, where some knowledge contained in the clients' model updates will be lost during aggregation $\sum\{w_{k,t} \mid k \in \mathbb{S}_t\} \to w_t$. This might be due to the coordinate-wise aggregation of weights as opposed to matched averaging in the parameter space of DNNs (Wang et al., 2020a).

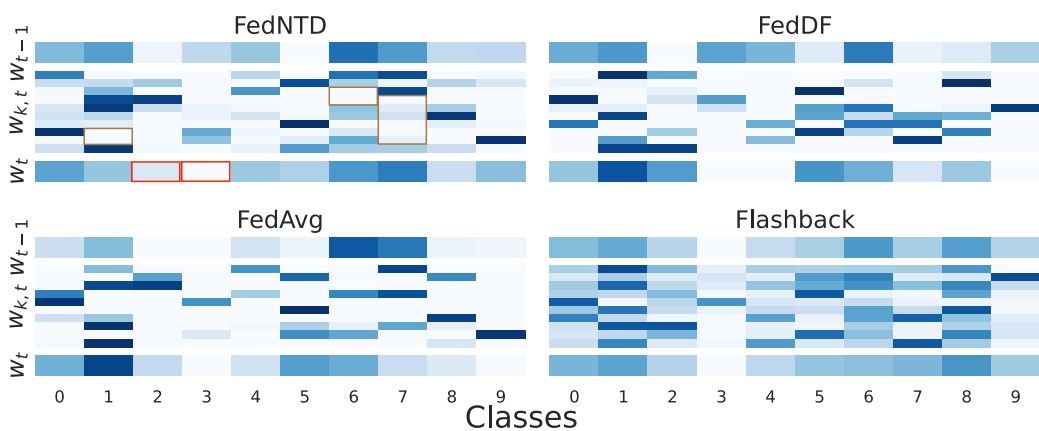

Figure 2: Local (client) & aggregation forgetting in some of the baselines using CIFAR10. The first row represents the global model per-class test accuracy at round $t-1$; then, the rows in the middle are all the clients that participated in round $t$, and finally, in the last row, the global model at the end of round $t$. Local forgetting happens when clients at round $t$ lose the knowledge that the global model had at round $t-1$ (example highlighted in brown). The aggregation forgetting happens when the global model at round $t$ loses the knowledge that in the clients' models at round $t$ (example highlighted in red).

We illustrate forgetting in Fig. 2 based on actual experiments with several baseline methods. Given round $t$, the figure shows the per-class accuracy of the global model $w_{t-1}$, all local models $w_{k,t}$, and the new global model $w_t$ for four different methods. The local forgetting is evident in the drop in accuracy (lighter shade of blue) of the local models $w_{k,t}$ compared to the global model $w_{t-1}$. The aggregation forgetting is evident in the drop in accuracy of the global model $w_t$ compared to the local models $w_{k,t}$. The figure also previews a result of our method, Flashback, which significantly mitigates forgetting. In summary, local and aggregation forgetting lead to the main forgetting problem in FL, which we term both as **round forgetting**, affecting $w_{t-1} \rightarrow w_t$.

In CL, forgetting is often quantified using Backward Transfer (BwT) (Chaudhry et al., 2018). FedNTD (Lee et al., 2021) adapted this metric, i.e., the forgetting score $\mathcal{F}$, for FL as follows:

$$\mathcal{F} = \tfrac{1}{C} \sum_{c=1}^{C} \arg\max_{t \in 1, T-1} (A_t^c - A_T^c), \qquad (2)$$

where $C$ is the number of classes and $A_t^c$ is the global model accuracy on class $c$ at round $t$.

However, $\mathcal{F}$ is a coarse-grain score that evaluates forgetting in aggregate across all rounds. We seek a finer-grain metric that measures forgetting round-by-round. Furthermore, we wish to account for knowledge replacement scenarios, such as when a decline in accuracy for one class might be accompanied by an increase in another, essentially masking the negative impact of forgetting in aggregate measures. Thus, for our evaluation results (§ 5), we propose to measure **round forgetting** by focusing only on drops in accuracy using the following metric:

$$\mathcal{F}_t = -\tfrac{1}{C} \sum_{c=1}^{C} \min(0, (A_t^c - A_{t-1}^c))$$

where $t > 1$ is the round at which forgetting is measured.

Our metric accounts for the pitfalls of the previously proposed forgetting metric. It discounts knowledge replacement scenarios that can happen between rounds by only focusing on the negative changes in accuracy. Furthermore, it provides a granular view of forgetting because it measures *round forgetting* (whereas Eq. (2) measures the global model forgetting at the end of training).

## 4 FLASHBACK: FORGETTING ROBUST FL

Our key idea to mitigate *round forgetting* is to leverage a dynamic form of knowledge distillation, which is fine-tuned in response to the evolving knowledge captured by the different models in the

---

**Algorithm 1** Flashback algorithm.

---

**input** Initial global model $w_0$, number of rounds $T$, fraction of clients $R$, minibatch size $B$, number
    of local epochs $E$, number of server epochs $E_s$, learning rate $\eta$
**output** global model $w_T$
1: $\boldsymbol{\pi} = \mathbf{0} \in \mathbb{R}^C$ // Global model's label count vector
2: **for** $t = 1$ **to** $T$ **do**
3:    $\mathbb{S}_t \leftarrow$ Randomly select $\lceil R \cdot N \rceil$ clients
4:    **for** each client $k \in \mathbb{S}_t$ **do**
5:       $w_{k,t} \leftarrow w_{t-1}$ // Initialize local model with current global model
6:       Compute $\boldsymbol{\alpha}$ with $\boldsymbol{\nu}$ as the local label count and a single teacher $\boldsymbol{\mu} \leftarrow \boldsymbol{\pi}$
7:       Update $w_{k,t}$ using dKD loss $\mathcal{L}_{\text{dKD}}$ **for** $E$ **epochs**
8:    **end for**
9:    $m_t \leftarrow \sum_{k \in \mathbb{S}_t} n_k$ // Total data points in this round
10:   $w_t \leftarrow \sum_{k \in \mathbb{S}_t} \frac{n_k}{m_t} w_{k,t}$ // Average to obtain the new global model
11:   $\mathbb{T} \leftarrow \{w_{k,t} \mid k \in \mathbb{S}_t\} \cup \{w_{t-1}\}$
12:   Compute $\boldsymbol{\alpha}$ with $\boldsymbol{\nu} \leftarrow \boldsymbol{\pi}$ and $\boldsymbol{\mu}_i$ as the label count $\forall w_i \in \mathbb{T}$
13:   Update $w_t$ using dKD loss $\mathcal{L}_{\text{dKD}}$ **for** $E_s$ **epochs**
14:   $r_k \leftarrow$ (Increment $r_k$ for every client $k \in \mathbb{S}_t$)
15:   **for** each client $k \in \mathbb{S}_t$ **do**
16:     **if** $\gamma r_k \leq 1$ **then**
17:       $\boldsymbol{\pi} \leftarrow \boldsymbol{\pi} + \gamma \boldsymbol{\mu}_k$ // Update participation count for client $k \in \mathbb{S}_t$
18:     **end if**
19:   **end for**
20: **end for**

---

training process. During local training, distillation ensures that each local model learns from the client's local dataset while retaining knowledge from the current global model. On the server side, after the clients' updates, Flashback begins by aggregating the locally updated models—much in the vein of FedAvg. Then, Flashback distills the knowledge of the freshly updated global model using our dynamic distillation approach, learning from both its immediate predecessor—the global model obtained at the previous round—and the ensemble of locally updated models, which all play the role of teachers. The Flashback algorithm is detailed in Algorithm 1. The remainder of this section discusses our distillation approach in more detail.

## 4.1 DYNAMIC DISTILLATION

As established in § 3, a client's local model can forget and override model knowledge with what is present in its private data. Moreover, the global model can be imperfect for two reasons: i) As we established before, the global model is susceptible to forgetting in the aggregation step. ii) Assuming no forgetting in the aggregation step, the knowledge contained in the clients who participated so far might not represent all the available knowledge, especially in the early rounds. Overall, both local models and the global model can be imperfect. Therefore, the logits of all the different classes cannot be treated equally (as in Eq. (1)), and the distillation loss has to adapt to the model's knowledge.

We propose using the label count to approximate the knowledge within a model. Here, the label count refers to the occurrences of each class in the training data that the model saw during training. In machine learning, a model's knowledge is fundamentally tied to the data it has been exposed to. If certain classes have higher representation (or label counts) in the training data, it's intuitive that the model would have more opportunities to learn the distinguishing features of such classes. Conversely, underrepresented classes might not offer the model sufficient exposure to learn their nuances effectively.

Our experimental results suggest that per-class performance on the test set correlates highly with the label counts in the training data. In scenarios where certain classes were more abundant, the model demonstrated higher proficiency in predicting those classes on the test set. As an example, Fig. 8 in the appendix illustrates for a randomly chosen client that the client's model performance on the test set well reflects the label count distribution of its private data. We conclude from this and many similar observations that the label count can indicate a model's knowledge. Furthermore, in

Appendix B we discuss a possible alternative to label count, which may be useful when label count can not be shared.

In standard knowledge distillation (Eq. (1)), all logits are treated equally since it is assumed that the teacher model has been trained on a balanced dataset. Owing to the heterogeneity of data distribution in local datasets, this assumption does not hold in FL. As a result, we cannot directly treat the current global model nor the local model updates as equally reliable teachers across all classes. Instead, we propose weighting the logits using the label count to approximate the per-class knowledge within a model.

We now revisit the distillation loss in Eq. (1) and transform the scalar $\alpha$ to a matrix form that is automatically tuned according to the label count of both the student and the teachers and used directly within the KL divergence loss. Namely, the dynamic $\boldsymbol{\alpha}$ parameter (defined below) will change during the training as the label counts change. Flashback maintains the global model counts over the rounds; this mechanism is detailed in the next section.

We consider a single student model $F_{w_s}$ with weights $w_s$ and a set $\mathbb{T}$ of $K$ teacher models; the $i$-th teacher model is denoted as $F_{w_i}$ with weights $w_i$. Let $\boldsymbol{\nu} \in \mathbb{R}^C$ be the relative label count vector of the student model, where $\nu^c$ is the relative occurrences of class $c$ in the dataset. Similarly, let $\boldsymbol{\mu}_i \in \mathbb{R}^C$ be the relative label count vector of the $i$-th teacher model.

The dynamic $\boldsymbol{\alpha} \in [0,1]^{K \times C}$ is defined as $[\boldsymbol{\alpha}_1^{\mathsf{T}}, \ldots, \boldsymbol{\alpha}_K^{\mathsf{T}}]$, with $\alpha_i^c = \frac{\mu_i^c}{\nu^c + \sum_k \mu_k^c}$.

Then, we embed $\boldsymbol{\alpha}$ directly in the KL divergence loss ($\mathcal{L}_{\text{KL}}$ in Eq. (1)) as follows:

$$\mathcal{L}_{\text{dKL}}(\boldsymbol{p}, \boldsymbol{q}; \boldsymbol{\alpha}_i) = \sum_{c=1}^{C} \alpha_i^c \cdot p^c \log\left(\frac{p^c}{q^c}\right)$$

Similar to standard distillation, to account for the student model knowledge with respect to the ground-truth class $y$, we define $\alpha_s^c = \frac{\nu^c}{\nu^c + \sum_k \mu_k^c}$. Thus, $\alpha_s^c + \sum_{k=1}^{K} \alpha_k^c = 1$ for all classes $c \in [C]$.

Finally, the dynamic knowledge distillation loss ($\mathcal{L}_{\text{dKD}}$) is:

$$\begin{aligned}
\mathcal{L}_{\text{dKD}}((x,y); w_s, \mathbb{T}, \boldsymbol{\alpha}) &= \alpha_s^y \mathcal{L}_{\text{CE}}(F_{w_s}(x), y) \\
&+ \sum_{w_i \in \mathbb{T}} \mathcal{L}_{\text{dKL}}(F_{w_i}(x), F_{w_s}(x); \boldsymbol{\alpha}_i)
\end{aligned} \tag{3}$$

The dynamic $\boldsymbol{\alpha}$ will weigh the divergence between the logits of different classes, making the student model focus more on learning from the teacher's strengths while being cautious of its weaknesses. This is important in FL because of the data heterogeneity problem. For instance, the global model may not encounter certain classes in the initial training rounds. Our distillation approach assigns a zero weight to the divergence of these classes, shielding the client model from adopting unreliable knowledge from the global model. Similarly, if a client possesses significantly larger data for a specific class than the global model has encountered, the weight assigned to that class's divergence will be small. This implies that the client's model remains more grounded in classes where it has more comprehensive data.

An interesting property of our distillation is that it will ignore the global model as a teacher in the first communication round. Since the initial global model label counts are all zeros, Eq. (3) reduces to just the cross-entropy: $\mathcal{L}_{\text{dKD}} = \mathcal{L}_{\text{CE}}$.

## 4.2 ESTIMATING THE GLOBAL MODEL KNOWLEDGE

Note that to apply the dynamic distillation loss Eq. (3), we must obtain the student's and teachers' label count vectors. While the label count of local models can be easily obtained (from the class frequency of local datasets), the label count of the global model is not readily available. We construct $\boldsymbol{\pi}$, the global model's relative label count, as follows. Let $r_k$ denote the number of rounds in which client $k$ participated in the training. For every client $k$ that participates at round $t$, Flashback adds a fraction $\gamma \in (0, 1]$ of client $k$'s label count ($\boldsymbol{\mu}_k$) to $\boldsymbol{\pi}$, unless $\gamma r_k > 1$, in which case $\boldsymbol{\pi}$ is not updated based on $k$'s label count. The latter case means that client $k$ has participated enough times that its label count is fully accounted for in $\boldsymbol{\pi}$.

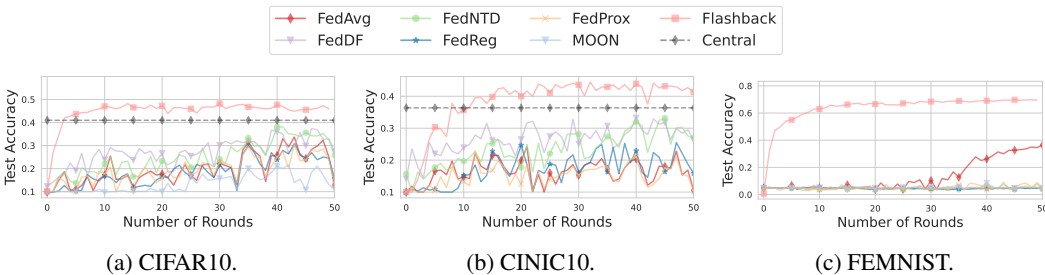

(a) CIFAR10.      (b) CINIC10.      (c) FEMNIST.

Figure 3: Round-to-accuracy performance of Flashback and other baselines over training rounds.

Intuitively, the parameter $\gamma$ indicates the rate at which we rely on the global model. When $\gamma$ is set to 1, it implies complete trust in the global model's ability to incorporate the clients' knowledge after just one round of participation. However, expecting such immediate and full assimilation is unrealistic, so we typically set $\gamma < 1$. The gradual build-up of the global label count plays a vital role in maintaining a balanced distillation in Eq. (3) during local updates. This progressive approach mirrors our growing trust in the global model's capabilities. It prevents the risk of assigning excessively high weights too soon, which could otherwise hurt the learning process.

## 5 EXPERIMENTS & RESULTS

We outline and analyze our experimental findings to investigate whether mitigating round forgetting successfully addresses the slow and unstable convergence issues due to forgetting problems as laid out in § 1. The experimental results stem from three settings: CIFAR10 (Krizhevsky, 2009) and CINIC10 (Darlow et al., 2018), where heterogeneous data partitions are created using Dirichlet distribution with $\beta = 0.1$ and FEMNIST (Caldas et al., 2019) with 3,432 clients, following the natural heterogeneity of the dataset. Furthermore, we do an ablation study on the different components of the algorithm. We use the same neural network architecture used in (Lee et al., 2021; McMahan et al., 2017), a 2-layer Convolutional Neural Network (CNN). Summaries of the datasets, partitions, and more details on the experimental setup are reported in Appendix D.

We compare Flashback against several baseline methods, namely: 1) FedAvg (McMahan et al., 2017), 2) FedDF (Lin et al., 2020), 3) FedNTD (Lee et al., 2021), 4) FedProx (Li et al., 2020), 5) FedReg (Xu et al., 2022), 6) MOON (Li et al., 2021b). It is noteworthy that both FedNTD and FedReg target forgetting in FL (discussed further in Appendix F). Flashback's server-side distillation is performed until early stopping is triggered on the public validation set (details in Appendix D). Moreover, Flashback only introduces one additional hyperparameter $\gamma$, representing how fast trust is built in the global model. We analyze the effect of $\gamma$ in Appendix E. We first evaluate Flashback performance by showing round-to-accuracy, round forgetting, and the local-global loss over the rounds. Moreover, we show additional results that explores various aspects of Flashback and show its soundness in Appendix E.

Table 1: Number of rounds to accuracy $A_x = A \cdot x$ where $A$ is the target accuracy and $x$ is a fraction.

| | CIFAR10, $A = 48.2\%$ | | | CINIC10, $A = 43.5\%$ | | | FEMNIST, $A = 69.5\%$ | | |
|---|---|---|---|---|---|---|---|---|---|
| | $A_{0.5}$ | $A_{0.75}$ | $A_{0.95}$ | $A_{0.5}$ | $A_{0.75}$ | $A_{0.95}$ | $A_{0.5}$ | $A_{0.75}$ | $A_{0.95}$ |
| FedAvg | 12 | 82 | - | 13 | - | - | 49 | 75 | 138 |
| FedDF | 7 | 40 | 112 | **2** | 30 | - | - | - | - |
| FedNTD | 12 | 41 | 112 | 13 | 46 | - | - | - | - |
| FedProx | 35 | 93 | - | 13 | - | - | 142 | - | - |
| FedReg | 35 | 108 | - | 16 | - | - | - | - | - |
| MOON | 82 | - | - | 124 | - | - | - | - | - |
| Flashback | **2** | **4** | **10** | 4 | **5** | **6** | **3** | **5** | **16** |

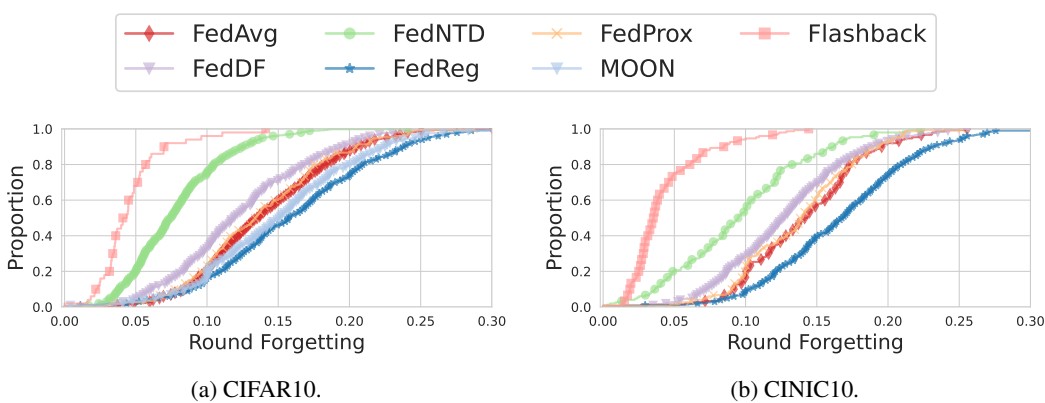

(a) CIFAR10.

(b) CINIC10.

Figure 4: Distribution of round forgetting of Flashback compared to other baselines.

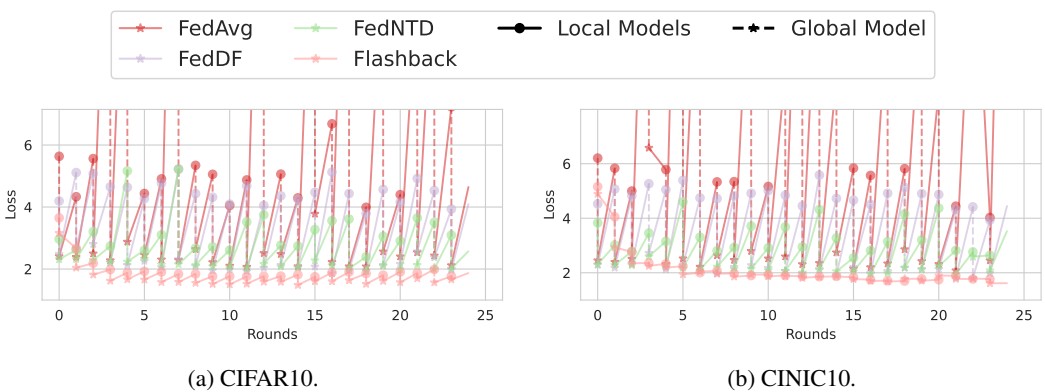

(a) CIFAR10.

(b) CINIC10.

Figure 5: Transition of local models loss to the global model loss over the rounds.

**Improved round-to-accuracy.** We evaluate the learning efficiency of Flashback and other baselines by showing the accuracy over rounds in Fig. 3; we include the result of central training on the public dataset. Flashback consistently shows faster convergence and high accuracy. Furthermore, we show the number of rounds it takes to reach a target accuracy and fractions of that target accuracy in Table 1. This indicates that addressing local forgetting and aggregation forgetting does provide training stability and, indeed, a faster convergence.

**Reduced round forgetting.** We show the empirical cumulative distribution function (ECDF) of the round forgetting in Fig. 4. We see that Flashback successfully reduces round forgetting. Also, FedNTD has less round forgetting than the remaining baselines. In the appendix, we show the round forgetting over the rounds (Fig. 14).

**Minimizing local models' divergence.** To further understand the effect of Flashback on the training behavior, we show the transition of the mean loss of the local models to the loss of the global model over the rounds in Fig. 5. This gives us an insight into the effect of the regularization made by our dynamic distillation. We see that the mean loss of the local models of the other baselines always spikes, signifying a divergence of these models from the global training objective, while Flashback has a much more stable loss. This shows that Eq. (3) regularizes the local models well such that they do not diverge too much from the global training objective.

In the following, we delve deeper into Flashback to understand its behavior and validate its performance gains.

**Training on the public dataset.** We conduct experiments to answer the following question: *does the performance improvement of Flashback come from the fact that we train the global model on a labeled public dataset?* To answer this question, we create a naïve baseline, where we extend FedAvg

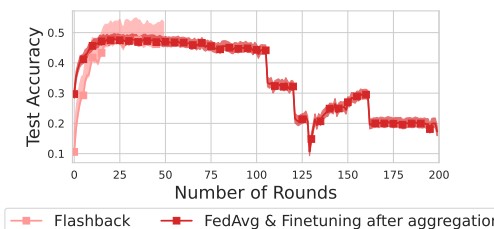 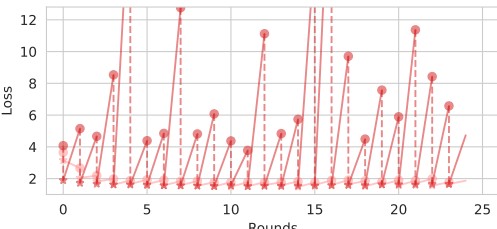

Figure 6: Comparing Flashback to FedAvg with fine-tuning: (left) test accuracy over 3 runs on CIFAR10; (right) the transition of the local models' loss to the global model loss over the rounds.

to fine-tune the global model using the labeled public dataset after the aggregation step at every communication round. From Fig. 6, we see that FedAvg with fine-tuning quickly reaches a stale model and eventually collapses. We believe this collapse happens due to the local models diverging too much after the local update such that the aggregation of those models would fail. This is evident by the spike of the local model's loss in the round-loss plot on the left.

## 6 RELATED WORK

**Federated learning.** FL is commonly viewed as an ML paradigm wherein a server distributes the training process on a set of decentralized participants that train a shared global model using local datasets that are never shared (Konečnỳ et al., 2015; Shokri & Shmatikov, 2015; Konečnỳ et al., 2016; Li et al., 2020; McMahan et al., 2017; Kairouz et al., 2021). FL has been used to enhance prediction quality for virtual keyboards among other applications (Bonawitz et al., 2019; Yang et al., 2018). A number of FL frameworks have facilitated research in this area (Caldas et al., 2019; OpenMined, 2020; tensorflow.org, 2020; Abdelmoniem et al., 2023).

**Heterogeneity in FL.** A key challenge in FL systems is uncertainties stemming from learner, system, and data heterogeneity. The non-IID distributions of Learners' data can significantly slow down convergence (McMahan et al., 2017; Kairouz et al., 2021) and several algorithms are proposed as means of mitigation (Fourati et al., 2023; Wang et al., 2020b; Karimireddy et al., 2020; Li et al., 2020; 2021a).

**Forgetting in FL.** Forgetting in FL has been explored in several studies, though many have limitations in addressing the full scope of the issue. We discuss in detail the main baselines that tackle forgetting in Appendix F. Luo, Kangyang, et al. (Luo et al., 2023) discuss forgetting due to local updates. Similarly, (Liu et al., 2022) tackles the problem of learning personalized models without forgetting what the global model has learned by using knowledge distillation. However, these approaches focus on the local update without addressing forgetting at the aggregation step. On the other hand, (Huang et al., 2022) focuses on domain shifts and clients with different data domains, aiming for personalized models rather than a global model. (Qu et al., 2022) investigate the convergence behavior and forgetting of Transformers compared to other architectures used in FL. Their experiments show that Transformers are robust to data heterogeneity. While these works address client heterogeneity, they do not delve into the forgetting issue in FL.

Overall, there is no investigation or exploration of what forgetting in FL entails. In Flashback, we provide a detailed analysis of forgetting in FL, demonstrating where and why it happens, and propose a metric to measure forgetting. Unlike previous works, which focus on client-side forgetting, Flashback addresses forgetting as a compound problem that occurs both at the local update and the aggregation step, suggesting it must be tackled at both levels.

## 7 CONCLUSION

We explored the phenomenon of forgetting in FL. Our investigation revealed that forgetting occurs during both the local update and the aggregation step of FL algorithms. We presented Flashback, a novel FL algorithm explicitly designed to counteract round forgetting by employing dynamic knowledge distillation. Our approach leverages data label counts as a proxy for knowledge, ensuring

a more targeted and effective forgetting mitigation. Our empirical results showed Flashback's efficacy in mitigating round forgetting, thereby supporting the hypothesis that the observed slow and unstable convergence in FL algorithms is closely linked to forgetting. This result underlines the importance of addressing forgetting, paving the way for advancing more robust and efficient FL algorithms.

**Flashback Limitation.** In this work, we focused on forgetting caused by data heterogeneity manifested by heterogeneity in the class distribution of clients. In the future, we look to investigate forgetting under other types of heterogeneity. Moreover, Flashback mainly uses label count to approximate knowledge, which could pose privacy challenges in certain scenarios. We discuss a possible alternative in Appendix B. As for computational overhead, Flashback adds 1 additional forward pass per iteration similar to FedNTD (Lee et al., 2021) and less than MOON (Li et al., 2021b), which adds 2 forward passes.

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

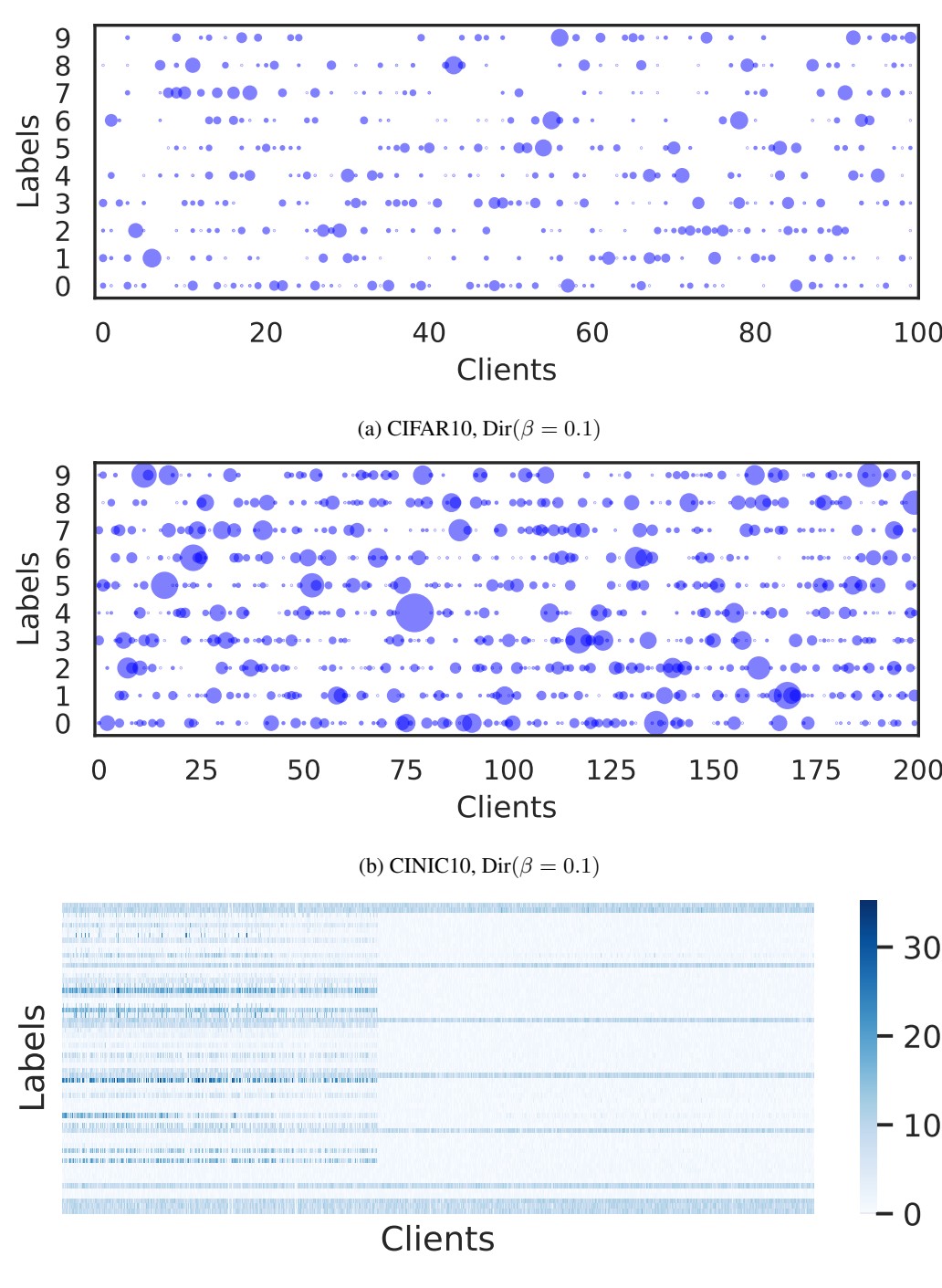

(a) CIFAR10, Dir($\beta = 0.1$)

(b) CINIC10, Dir($\beta = 0.1$)

(c) FEMNIST with 3432 clients

Figure 7: Clients data distribution. The x-axis is the clients and the y-axis is the labels.

## A LABEL COUNT MOTIVATION

Fig. 8 shows, on the left, the per-class accuracy of a randomly-chosen example client from a FedAvg training experiment. On the right, the figure shows the corresponding label count at the client. This example suggests that label count can be representative of the model performance.

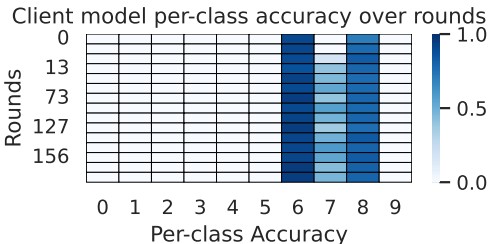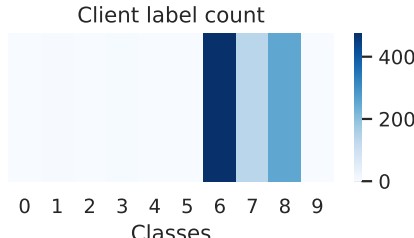

Figure 8: (left) Per-class accuracy of a client model on all the rounds where it participated. (right) Data distribution of that client.

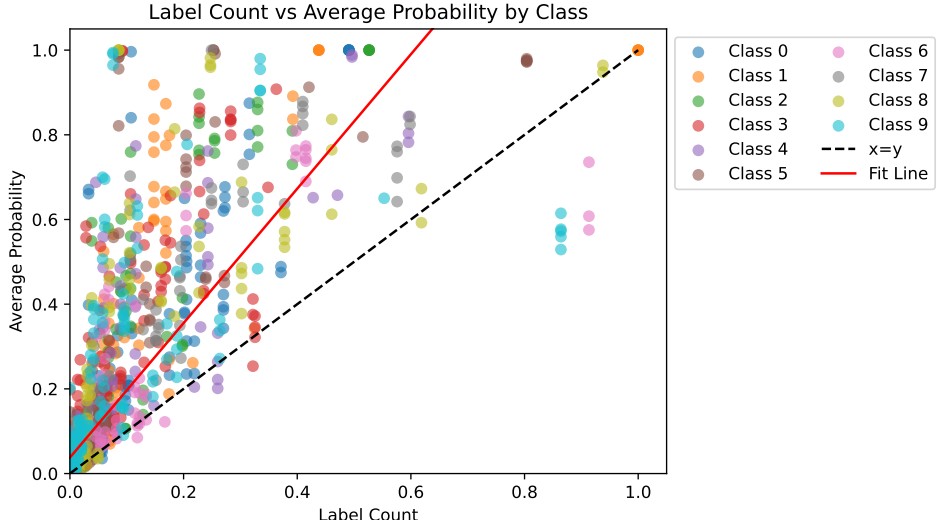

Figure 9: Label count of a class normalized by the maximum count of the given class, versus the average probability of a class obtained via a client model after a local update.

## B  KNOWLEDGE APPROXIMATION

Flashback uses the label count of clients as a simple way to approximate the knowledge of the client's models. However, using the label count may raise privacy concerns. In this section, we discuss another possible way of approximating knowledge.

Similar to the label count, intuitively the average output probability (logits) of a model can be used to approximate its learned knowledge. To validate this conjecture, we conduct a simple experiment to look at how similar the average probabilities of the client's models –obtained via FedAvg– to their respective label count. Specifically, we computed the logits of all the models obtained after each client update in the first 30 rounds on the public training set. In Fig. 9 we observe a correlation between the average probability of the classes and their count. However, the intensity of weights given by average probabilities will be different than the label count. As a result, the $\gamma$ parameter of Flashback needs to be tweaked accordingly.

## C  FORGETTING IN FEDERATED LEARNING AND CONTINUAL LEARNING

Forgetting is a prominent problem in CL, where tasks change over time, and consequently, the data distribution, places models at risk of overriding previously learned knowledge. Looking at FL with the same perspective, we have the intra-round and inter-round data distribution heterogeneity. Intra-round clients with different data distributions participate, by updating the current global model weights $w_{t-1}$. The server obtains a set of $\{w_{k,t} \mid k \in S_t\}$ where $S_t$ is the set of clients participating

in round $t$. The goal is after the aggregation step, a new global model $w_t$ is obtained containing all the knowledge that was present in $\{w_{k,t} \mid k \in S_t\}$. Inter-round the global model $w_t$ has learned knowledge that over the prior rounds $1 \ldots t$. At each round, different sets of clients participate. The goal is to carry this knowledge to the next round $t + 1$ even though the new set $S_{t+1}$ most likely is different with respect to the data distribution than the previous set $S_t$. Presenting distinct forgetting challenges in FL compared to CL.

# D    EXPERIMENTS DETAILS

## D.1    DATASETS

In this section, we provide an overview of the datasets used, the data split, and the specific experimental setups. For each dataset, we perform two sets of experiments to analyze the effects of data heterogeneity on the algorithms' performance. The datasets used are CIFAR10, CINIC10, and FEMNIST.

**CIFAR10** (Krizhevsky, 2009). A famous vision dataset that includes 50k training images and 10k testing images. We emulate a realistic, heterogeneous data distribution by using a Dirichlet distribution with $\beta = 0.1$. A $\beta$ value of 0.1 is chosen to simulate a more heterogeneous, and challenging data distribution. A 2.5% random sample of the training set creates a public dataset, further divided into training and validation sets. This yields a very small public training dataset with size of 1.88%. The remaining 97.5% is distributed among 100 clients, with each client's data being split into training (90%) and validation (10%) subsets.

**CINIC10** (Darlow et al., 2018). A drop-in replacement of CIFAR10, this dataset comprises 90k training, 90k validation, and 90k test images. We merge the training and validation sets and adopt a similar approach as with CIFAR10, taking out 2.5% of the data to be the public dataset, similar to the CIFAR10 case this 2.5% is further divided into training set and validation set. Then employing Dirichlet distribution with $\beta$ value of 0.1 to split the 97.5% remaining data into 200 clients, with each client's data further divided into training (90%) and validation (10%) sets.

**FEMNIST** (Caldas et al., 2019). This federated learning dataset is based on extended MNIST with natural heterogeneity, where each writer is considered a client. From the 3597 total writers, those with less than 50 samples are excluded. We randomly selected 150 writers to form a public dataset. The remaining 3432 writers' data is divided into train (approx. 70%), validation (approx. 15%), and test (approx. 15%) sets. The collective test sets from all writers form the overall test set. At every round, 32 clients are randomly selected for participation.

For CIFAR10 and CINIC10, we chose $\beta$ values of 0.1 and client participation value of 10. While for FEMNIST we have 3432 clients (writers) with client participation vale of 32 In all cases, the training data distribution among clients is illustrated in Fig. 7.

## D.2    BASELINES & HYPERPARAMETERS

We evaluate the following algorithms as baselines: 1. FedAvg (McMahan et al., 2017); 2. FedDF (Lin et al., 2020); 3. FedNTD (Lee et al., 2021); 4. FedProx (Li et al., 2020); 5. FedReg (Xu et al., 2022); and 6. MOON (Li et al., 2021b) . Both FedNTD and FedReg target forgetting in FL (discussed in § 6). We use the same neural network architecture that is used in (Lee et al., 2021; McMahan et al., 2017), which is a 2-layer CNN. Note that for MOON (Li et al., 2021b), we add an additional layer to the model as described in their source code for the projection head. Moreover, for the optimizer, learning rate, and model we follow (Lee et al., 2021; McMahan et al., 2017), and when a baseline has different hyperparameters we use their proposed values. For example, in FedDF the number of local epochs is set to 40, while in the other baselines and Flashback, it is set to 5 epochs. As for Flashback hyperparameters, during the server distillation, we train until early stopping gets triggered using the validation set; we set the label count fraction $\gamma = 0.025$ for CIFAR10, i.e., we add 2.5% of the client label count each time it participates, while we set $\gamma = 0.1$ for CINIC10 and FEMNIST. As for distillation-specific hyperparameters, we have one fewer hyperparameter since $\alpha$ is computed automatically, and for temperature, we use the standard $T = 3$.

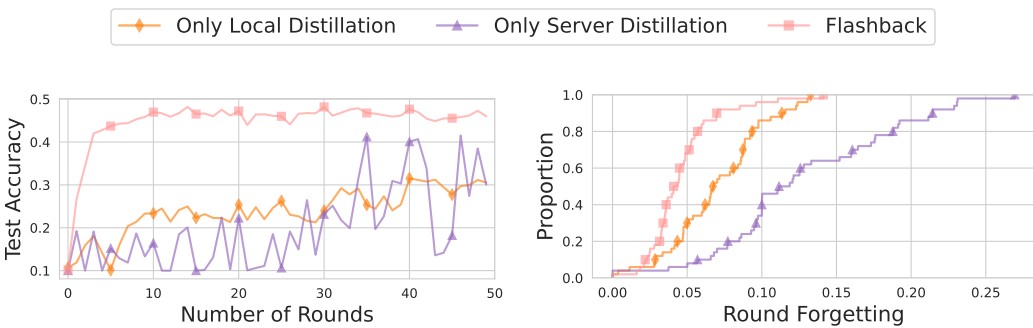

Figure 10: Performing distillation at only one side of the algorithm (client & server) on CIFAR10.

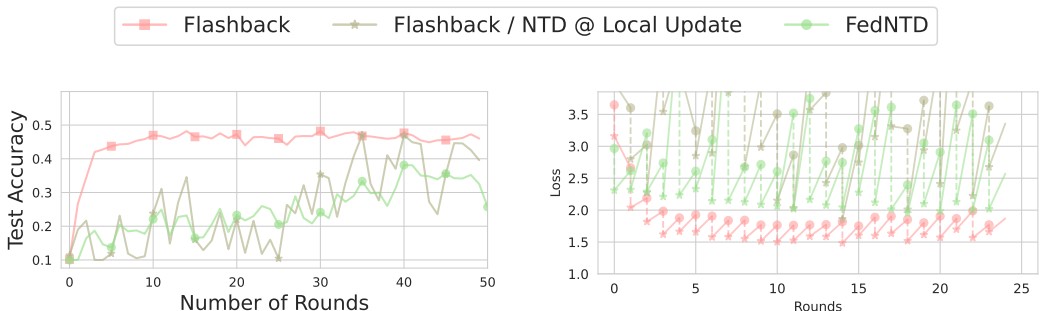

Figure 11: Using NTD loss instead of Flashback's dynamic distillation at the local update on CIFAR10.

# E  ADDITIONAL RESULTS

**Dissecting the distillation.** To show the importance of performing dynamic distillation during clients' updates and at the server's aggregation step, we conduct an experiment where we run Flashback with local distillation only and with server distillation only (c.f. Fig. 10). We observe that doing dynamic distillation at either side of the algorithm – client update and aggregation step – doesn't address forgetting or gets a similar performance to Flashback.

To validate the *importance of dynamic distillation at client- and server sides* towards Flashback's performance gains, we create a baseline where we replace the local distillation loss with not-true distillation (NTD) loss (Lee et al., 2021). From Fig. 11, we observe that this baseline doesn't perform as well as Flashback, and performs similarly to FedNTD. This indicates that performing the distillation at the server needs well-regularized local models (teachers), which is further supported by our previous experiment contrasting Flashback to single-side distillation (c.f. Fig. 10).

**Effect of public dataset size.** Flashback requires the availability of a public labeled dataset. This may be a limiting assumption in some cases. To study this limitation, we explore a few scenarios for the size of the public dataset in Fig. 12: 1) 9000 samples (15% of CIFAR10), 2) 1125 samples (1.88% of CIFAR10), 3) 1283 samples that have unbalanced class distribution (2.14% of CIFAR10), 4) 450 samples (0.75% of CIFAR10). For all of these scenarios, we train a model centrally on the public dataset. We find that Flashback can benefit from a large balanced public dataset. Most importantly, Flashback can work well with a small public dataset (1125 samples is the default in all experiments). Furthermore, even if the public dataset has a class imbalance Flashback still performs relatively well. In all of the cases, Flashback always outperforms central training on the public dataset. Overall, Flashback requires the availability of a public dataset, however, it does not require a huge amount of data or hard requirements for the class distribution to be very balanced.

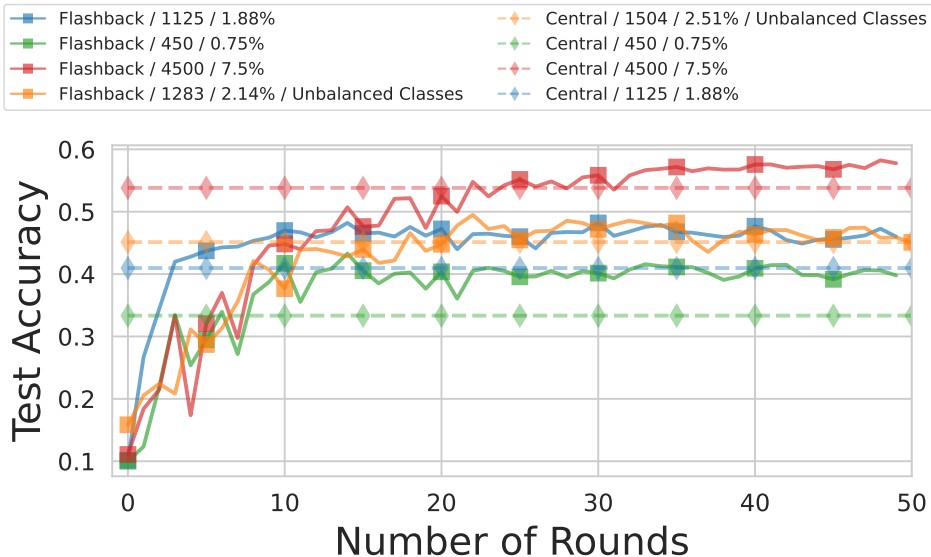

Figure 12: Flashback using different public datasets, and the results of central training on public datasets of different sizes.

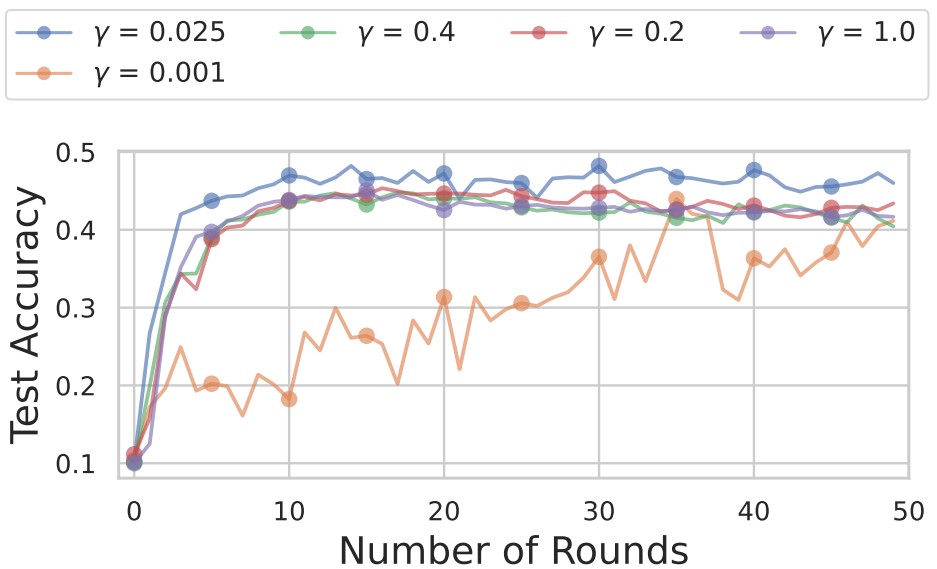

Figure 13: Varying the trust of the global model parameter $\gamma$.

**The importance of** $\gamma$. As mentioned, Flashback has a single hyperparameter $\gamma$, which dictates how fast models will trust the global model as a competent teacher. We explore the effect of this hyperparameter in Fig. 13. We find that setting this parameter to a larger value leads the learning process to get to a stale solution quickly. This is intuitive since large $\gamma$ leads the global model label count to grow faster, therefore, this dominates the loss term in Eq. (3) during both the client- and the server update. Smaller $\gamma$ gives the best performance because it gives the local models time to learn from their own private data by having a small weight to the distillation term in Eq. (3). However, too small value for $\gamma$ such as $0.001$ slows the training process, since during the local training the distillation term in Eq. (3) will be very small in the early rounds.

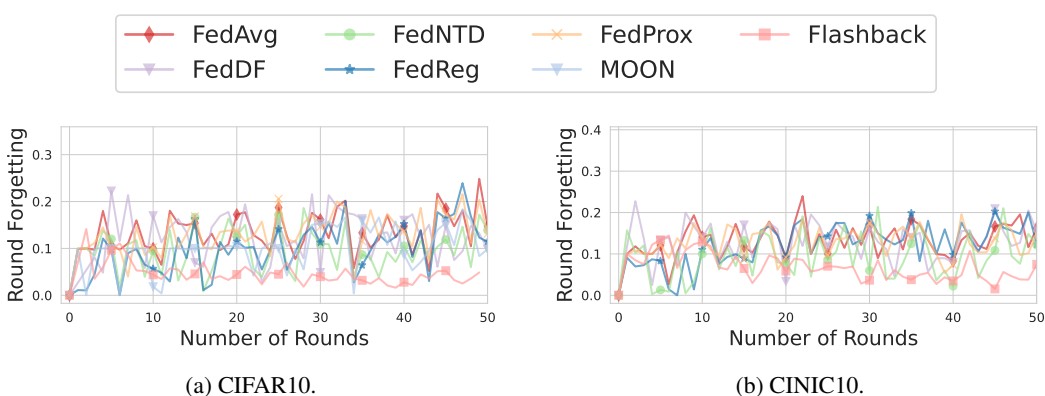

(a) CIFAR10.    (b) CINIC10.

Figure 14: The round forgetting of Flashback and other baselines over training rounds.

In Fig. 14, we show the round forgetting score computed over the rounds. We see that the baselines have very flaunting round forgetting score.

In Fig. 15, we show the transition of the average loss of the local models to the global model loss on the test set. Interestingly, we can see that even though performing local distillation only doesn't have the same performance as Flashback, it does mitigate the local forgetting. That is, we do not see a spike in the loss after the clients perform their local update.

In Fig. 16, we see the ECDF of the round forgetting. The Flashback variant with NTD even shows worse forgetting than Flashback and FedNTD. Further showing that just performing distillation at both sides doesn't address the forgetting problem.

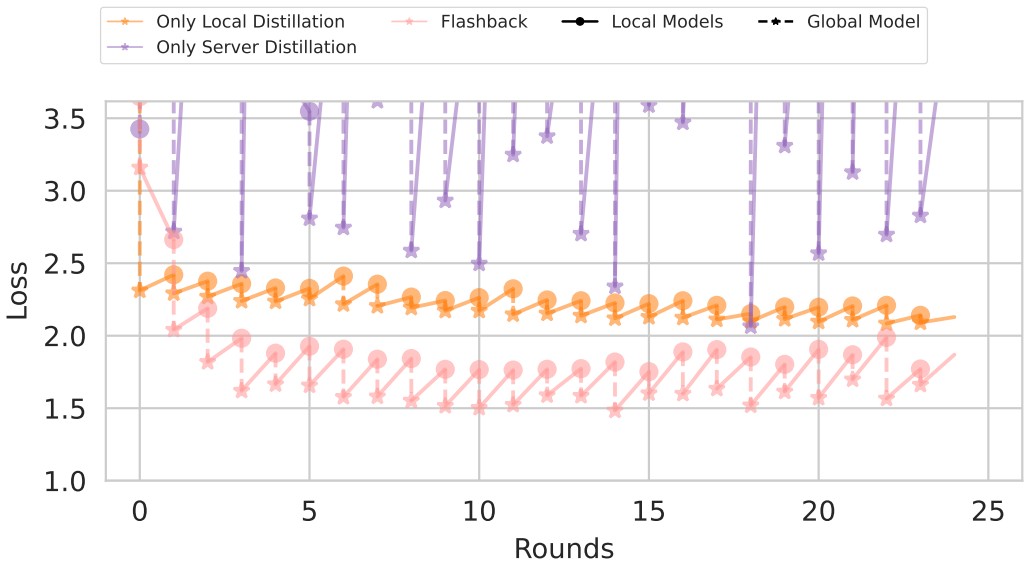

Figure 15: Performing distillation at only one side of the algorithm (client & server) on CIFAR10.

## F    EXTENDED RELATED WORK

This section delves deeper into the main forgetting baselines we compare wit FedReg (Xu et al., 2022) and FedNTD (Lee et al., 2021).

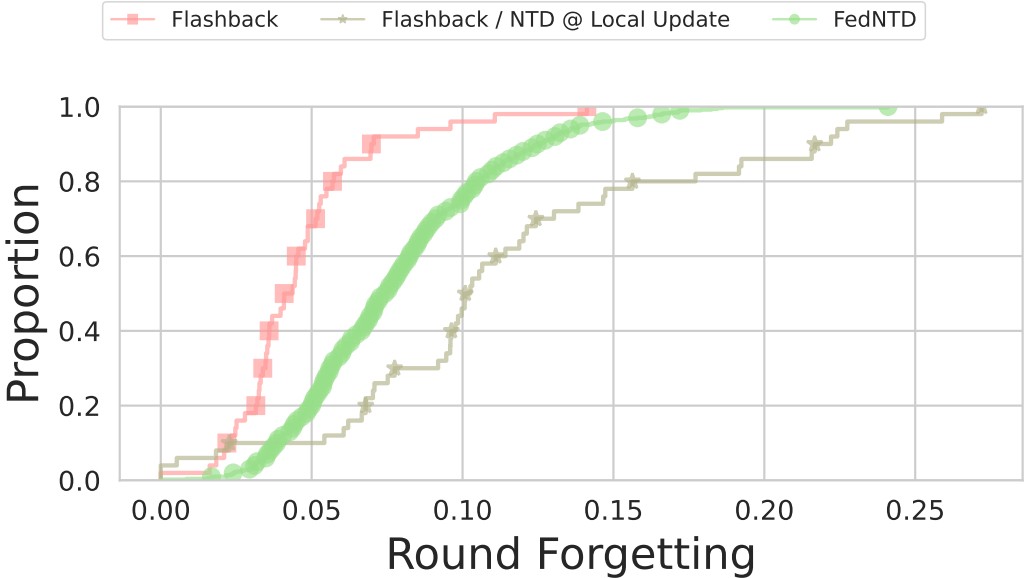

Figure 16: Using Not-True Distillation instead of Flashback's dynamic distillation at the local update on CIFAR10.

**FedReg** (Xu et al., 2022) addresses the issue of slow convergence in FL, asserting it to be a result of forgetting at the local update phase. They demonstrate this by comparing the loss of the global model $w_{t-1}$ on specific client data points with the averaged loss of updated clients' models $\{w_{t,k} \mid k \in \mathbb{S}_t\}$ on the same data points, highlighting a significant increase in the average loss, indicative of forgetting. However, our work proposes a systematic way of measuring forgetting using a metric designed to capture it. Furthermore, we show that forgetting doesn't only occur in the local update and at the aggregation step (§ 3 and Fig. 2). FedReg proposes to generate fake data that carries the previously attained knowledge. During the local update, Fast Gradient Sign Method (Goodfellow et al., 2014) is used to generate these data using the global model $w_{t-1}$ and the client data. Then, the loss of the generated data is used to regularize the local update. While FedReg employs regularization using synthetic data during local updates, our work, Flashback, leverages dynamic distillation to ensure knowledge retention at both local updates and aggregation steps.

**FedNTD** (Lee et al., 2021) makes a connection between CL and FL, suggesting that forgetting happens in FL as well. Similarly to FedReg, their analysis shows that forgetting happens at the local update, where global knowledge outside of the client's local distribution is susceptible to forgetting. To address this, they propose to use a new variant of distillation Eq. (1) named Not-True Distillation (NTD), that masks the ground-truth class logits in the KL divergence as $\mathcal{L}_{\text{KL}}(\boldsymbol{p}, \boldsymbol{q}) = \sum_{i=c, c \neq y}^{C} p^c \log(\frac{p^c}{q^c})$, where $y$ is the ground-truth class. NTD is used at the local update, while all the other steps in the algorithm remain the same as FedAvg. FedNTD aims to preserve global knowledge during local updates.

Both FedReg and FedNTD diagnose the issue of forgetting primarily within the realm of local updates, asserting that this stage risks losing valuable global knowledge. Consequently, both works present innovative solutions specifically tailored to counteract this local update forgetting. However, their perspective overlooks a pivotal aspect of the forgetting problem: the occurrence of forgetting during the aggregation step. As we delve into in § 3, this oversight in recognizing and addressing forgetting during aggregation has repercussions on the later local updates. In contrast, Flashback takes a holistic approach, comprehensively targeting forgetting across the local updates and the aggregation phase, leading to faster convergence.

