# OpenReview forum: "Flashback: Understanding and Mitigating Forgetting in Federated Learning"
_ICLR.cc/2025/Conference — ICLR 2025 Conference Withdrawn Submission_

### Official Review · Reviewer_JksB · 2024-11-01

**Soundness:** 3
**Presentation:** 3
**Contribution:** 2
**Rating:** 5
**Confidence:** 4

**Summary:**

The motivation of this paper is to address the problem of forgetting in Federated Learning (FL), which slows down convergence, particularly in settings with high data heterogeneity among clients. The main contribution is the introduction of Flashback, a new FL algorithm that uses dynamic distillation to mitigate forgetting by regularizing both client-local updates and server aggregation.

**Strengths:**

(a) The method includes comparisons with a variety of baselines that incorporate regularization and distillation techniques.

(b) The paper is well-written, with clear presentation and structure that is easy to follow.

**Weaknesses:**

(a) The investigation of forgetting is less systematic than claimed. While the paper frames forgetting as a key factor in FL underperformance, it lacks detailed analysis and comparisons with baselines from continual learning, where many regularization- [1] based methods effectively mitigate forgetting. Exploring whether these methods can similarly address forgetting in FL would be valuable.

(b) Fairness in performance comparison on public datasets is an issue, as the proposed method uses a portion of data for validation while other methods do not, making the comparisons unfair. Including simple baselines that utilize similar validation strategies, such as server-side distillation [2] or model selection, would improve fairness.

(c) The motivation and application of the new forgetting metric in FL are not clearly explained, making it difficult to fully understand this contribution’s impact.

[1] Kirkpatrick, J., et al. “Overcoming catastrophic forgetting in neural networks.” Proceedings of the National Academy of Sciences 114.13 (2017): 3521-3526.

[2] Huang, Chun-Yin, et al. "Overcoming Data and Model Heterogeneities in Decentralized Federated Learning via Synthetic Anchors." arXiv preprint arXiv:2405.11525 (2024).

**Questions:**

(a) Why are so many values missing in Table 1, particularly for the FEMNIST dataset?

(b) Why is the baseline performance on FEMNIST so low, appearing to learn almost nothing, and why does performance fluctuate significantly across all datasets? This fluctuation suggests that hyperparameters may not be well-tuned for optimal convergence. Providing the training and validation loss trends for each baseline would help demonstrate convergence.

---

> ### Author Response · Authors · 2024-11-27
>
> We thank the reviewer for the feedback.
>
> a. This is a great point, we didn't explore baselines from CL because the setup is different. And to the best of our knowledge baselines wouldn't be directly applicable.
>
> b. This is a great point, and indeed we did that in the paper. Firstly FedDF is a method that uses the public dataset, and we show that Flashback performs better. Secondly, we create a baseline where FedAvg uses the public dataset to fine-tune after each round. In both cases, Flashback performed better.
>
> c. In the paper we establish that forgetting is a problem in FL, the metric should help researchers and practitioners see if forgetting occurs during the training.
>
> Questions:
> 1. It means that the methods weren't able to reach those accuracies.
> 2. FEMNIST is a harder dataset, and the other methods will require many more rounds to converge. As for the fluctuation we use the same hyperparameters for all the methods, and Flashback doesn't face any. And we claim that the cause of this fluctuation is the forgetting. We do provide something similar, in figure 5 we show the training loss of the global model and the average of the local models.

---

> > ### Comment · Reviewer_JksB · 2024-11-28
> >
> > Thank you for your response. My concerns regarding the baseline comparison and the clarity of the paper’s motivation remain. I will maintain my current score for this version of the paper.

---

### Official Review · Reviewer_fsgu · 2024-11-02

**Soundness:** 2
**Presentation:** 2
**Contribution:** 2
**Rating:** 3
**Confidence:** 4

**Summary:**

This paper proposes a new distillation method to solve the NonIID problem in FL. Specifically, they consider the NonIID problem will cause the forgetting of model knowledge in both local update and global aggregation processes. Then, they apply the distillation over both local and global processes to mitigate the forggeting.

**Strengths:**

1. The NonIID problem in FL is important and using distillation to solve this issue achieves promising results.

2. The writing is easy to follow.

**Weaknesses:**

1. The novelty is limited. In fact, there have been massive methods using knowledge distillation to solve the NonIID problem, e.g., [1,2]. The experiments should also include them for the comprehensiveness of comparison.

[1] DaFKD: Domain-Aware Federated Knowledge Distillation. CVPR 2023.

[2]  Data-free knowledge distillation for heterogeneous federated learning. ICML 2021.

2. It is better to include a figure to illustrate the method framework for ease of understanding.

3. It would be better if the theoretical advantages were provided.

4. The ablation study about the proposed method of local and global distillation should be provided.

**Questions:**

See above.

---

> ### Author Response · Authors · 2024-11-27
>
> We thank the reviewer for the feedback.
>
> 1. We don't claim that the use of distillation in FL is our novelty, we show that there is a forgetting problem in non-IID scenarios and provide a detailed analysis of it, we show that other baselines that targeted this problem didn't completely address forgetting. We design a new novel algorithm and new distillation method specifically to mitigate forgetting. We kindly disagree with the reviewer that our novelty is limited. We thank the reviewer for the suggested baselines, we believe that we still covered a good selection of baselines to compare against.
> 2. We thank the reviewer for the suggestion, but can the reviewer kindly clarify how is this a weakness?
> 3. We thank the reviewer for the suggestion, and we agree that having theoretical results would be advantages. However, we believe the contributions provided in the paper are still sufficient.
> 4. It is provided (figure 11) and discussed, so this weakness is false.

---

### Official Review · Reviewer_gsSF · 2024-11-05

**Soundness:** 2
**Presentation:** 2
**Contribution:** 2
**Rating:** 5
**Confidence:** 5

**Summary:**

his paper analyzes the problem of forgetting in a non.iid federated learning environment, and concludes that the forgetting occurs in local updating and aggregation processes. For this, the authors propose metrics to measure the degrees of forgetting in the training process. In addition, the authors propose Flashback, a knowledge distillation (KD)-based model to mitigate forgetting. Flashback allows both clients and the server to perform dynamic KD according to the relative label count, and achieve fast convergence against baselines.

**Strengths:**

1. The authors provide a detailed analysis and illustration of the key issues - where forgetting occurs in non.iid FL, which is an important problem in FL.

2. The proposed Flashback method is simple and easy to understand, and it outperforms a variety of FL baselines

3. This paper introduces fine-grained evaluation metrics for forgetting in FL.

**Weaknesses:**

1. This paper proposes a fine-grained metric for assessing forgetting, but this does not seem to be reflected in Flashback. The authors should emphasize the connection between this metric and Flashback.

2. I have concerns about the use of the public dataset on the server. If the distribution of that dataset is similar to the distribution of data on each client, does this mean that there is already a leakage problem in that environment? Also, I'm curious what happens if a different dataset is used than the training dataset (e.g., cifar10 as the public dataset and cinic10 as the training data).

3. The authors should reorganize Sections 3, 4 for clearer understanding, and placing the algorithm in 4.1 would make it easier to understand.

4. Flashback involves a variety of additional parameters (e.g., label counts for teacher and student models, dependency on the global model, etc.), do these parameters make the algorithm less robust to some extent? This paper would benefit from an analysis of this problem.

5. Missing some related works, e.g. test-time FL [1-3].

[1] L Jiang, T Lin, Test-Time Robust Personalization for Federated Learning, ICLR 2023
[2] Y Tan, et al., ​​Is heterogeneity notorious? taming heterogeneity to handle test-time shift in federated learning, NeurIPS 2023
[3] W Bao, et al., Adaptive Test-Time Personalization for Federated Learning. NeurIPS 2023

**Questions:**

In addition to the weaknesses, one more question is as below.

1. In the experiments, do all baselines use the public dataset on the server?

---

> ### Author Response · Authors · 2024-11-27
>
> We thank the reviewer for the feedback.
>
> 1. The metric should be agnostic to the method we proposed. The metric did help us come up with the design of Flashback. However we believe that having a connection between Flashback and the proposed metric is  **not** necessary, as the goal of the metric is to help identify if forgetting does happen during the training. Can the reviewer kindly elaborate more on why this point is considered a weakness of our work?
> 2. The public dataset does not need to have the same distribution of the clients. In our experiments, the public dataset didn't have the same distribution as the clients. So this weakness is not accurate.
> 3. We thank the reviewer for the suggestion. But can the reviewer clarify how is this considered a weakness of our contribution?
> 4. The only additional hyperparameter by Flashback is $\gamma$ which we report an experiment on it is effect in the appendix. The label count is not a parameter. This weakness is false, as we don't have many parameters we only have one, and we do show Flashback robustness to this hyperparameter (figure 13).
> 5. We kindly disagree as these works are not related to ours. As they tackle a different problem formulation where test sets evolve over time.
>
> Questions:
> 1. Only FedDF requires a public dataset, however, we made sure all the algorithms are evaluated with the same data split. Moreover, in figure 6, we create a baseline where FedAvg uses the public dataset, and it doesn't perform well.

---

### Official Review · Reviewer_uokM · 2024-11-05

**Soundness:** 2
**Presentation:** 2
**Contribution:** 1
**Rating:** 3
**Confidence:** 5

**Summary:**

This work focuses on addressing the problem of data heterogeneity in federated learning. This work analyzes the mechanism of data heterogeneity in multiple rounds of iterations of federated learning from the novel perspective of forgetting. In addition, this paper proposes a new metric for measuring the degree of forgetting between training rounds. The major contribution of this paper is the proposed Flashback algorithm, which utilizes knowledge distillation to mitigate forgetting during the local update and global update phases. In conclusion, this paper presents a novel perspective to study data heterogeneity in federated learning and proposes methodological solutions, and the article ideas are easy to understand.

**Strengths:**

This paper deals with data heterogeneity in federated learning from the perspective of forgetting which is very novel and gives a comprehensive framework from observation, derivation, solution design and experimental proof.

It is well written and easy to understand.

**Weaknesses:**

1 The researched work related to data heterogeneity in federated learning is insufficient, especially for personalized federated learning and clustered federated learning.

2 The elaboration of the concept of forgetting is not detailed enough, lacking comparisons between different categories of continual learning (class-CL, task-CL, domain-CL), and lacks a detailed elaboration of forgetting mechanisms (e.g., weight drift, activation drift, inter-task confusion, and task-recency bias).

3 The forgetting metric between training rounds proposed in this paper is also computed based on accuracy, which is not different enough from the forgetting rate metric in continual learning to be considered as a contribution point.

4 The idea of using knowledge distillation in Flashback has been widely used, and the approach in this paper lacks innovation.

5 Flashback's dependence on labeled datasets on the server side is the major limitation and is not feasible in real scenarios, perhaps try unlabeled datasets or data-free distillation.

6 The experiments are less persuasive:

The datasets used (CIFAR10, CINIC10, FEMNIST) are too simple, it would be more persuasive to use more complex datasets such as CIFAR100 or TinyImageNet.

Comparison methodology lacks the most recent work (including 2023 and 2024) and work related to personalized federated learning.

7 The font sizes in images is inconsistent, e.g., Figures 4 and 5.

**Questions:**

What are the differences between forgetting in federated learning and forgetting in continual learning?

What are the differences between the federated continual learning scenario and the federated learning scenario in this paper?

Since the paper states that forgetting occurs in both local updating and global aggregation phases, shouldn't there be more detailed forgetting metrics (e.g., phased forgetting metrics)?

---

> ### Author Response · Authors · 2024-11-27
>
> We thank the reviewer for the feedback.
>
> 1. Personalized federated learning and clustered federated learning are not within the scope of our work, it is not clear what is insufficient, can the reviewer kindly clarify this weakness?
> 2. There may be a misunderstanding by the reviewer as we are not working in a continual learning setup. Moreover, we believe that we clearly explained the kind of forgetting that is unique to the standard FL setup.
> 3. We proposed a forgetting metric that is adapted to the characteristics of the FL setup, maybe it is a minor contribution but we believe it is a contribution nonetheless.
> 4. We are proposing a new FL algorithm that uses a new kind of distillation tailored to address forgetting, the algorithm is novel as well as the distillation technique. Yes, many previous works used distillation in FL but that doesn't negate our new contributions. We kindly disagree with the reviewer.
> 5. Indeed Flashback requires a labeled dataset, but in our experiment, we show that it needs a very small labeled dataset. Any entity capable of running an FL training, it shouldn't be very difficult to obtain such a small dataset. Nonetheless, we acknowledge that is is a limitation of our work and not all the cases it is possible to get a public labeled dataset.
> 6. Personalized FL algorithms are not within the scope of our work, we disagree with the reviewer as CINIC10 is a complex dataset that  is much larger than CIFAR10 and CIFAR100, and has images from ImageNet.
> 7. Thanks for pointing this out.
>
> Questions:
> 1. FCL has completely different problem formulation. In FCL the Continual Learning problem formulation is adopted in FL where data changes over time (like CL). We show that in standard FL formulation, there is a forgetting problem.
> 2. In FCL data changes over time and previous data may never seen again, exactly like CL. However, in our setup, we follow standard FL setup, we do not change the problem formulation.
> 3. That is a great question, we came up with a variant of the metric that focused on the local update, but we opted not to include it as it wasn't very informative.

---

### Official Review · Reviewer_LHg4 · 2024-11-10

**Soundness:** 2
**Presentation:** 3
**Contribution:** 2
**Rating:** 3
**Confidence:** 4

**Summary:**

This paper formulates the NonIID problem in FL as the catastrophic forgetting from both local and global perspective. To solve this problem, they propose using knowledge distillation in both local and global sides to maintain knowledge to mitigate forgetting.

**Strengths:**

1.	The NonIID problem in FL is important and using distillation to solve this issue achieves promising results.
2.	The writing is easy to follow.

**Weaknesses:**

The major concern is that the experiments are too limited:
1.	All baselines are too old. The newest baseline FedReg which is proposed in 2022. In fact, many recent baselines for soving the NonIID problem are proposed in 2023 and 2024.
2.	The evaluations should be conducted on different NonIID settings. This paper only adopt a fixed NonIID setting across all datasets, which is not sufficient to demonstrate the effectiveness of the proposed method.
3.	Larger models such as ResNet18 can be included. Merely 2-layer CNN may not be enough.
4.	Varying fraction of one-round selected clients should be evaluated. This factor is strongly related to the global knowledge forgetting.
5.	Current datasets only cover 10-classes settings, which may be limited to make a comprehensive comparison because $\alpha$ proposed by the method is strongly related to the number of classes. Cifar100 is recommended to add to the experiments.

**Questions:**

1.	Federated Continual Learning (FCL) also focuses on the catastrophic forgetting problem. What’s the difference between the problem of this paper and that of FCL?

---

> ### Author Response · Authors · 2024-11-27
>
> We thank the reviewer for the feedback.
>
> 1. We mainly focused on baselines that target the problem of forgetting in FL to show that they didn't completely address the problem and there is room for improvement.
> 2. This is false as we show results for two different non-IID settings, one generated through Dirichlet distribution and a natural non-IID distribution of the FEMNIST dataset.
> 3. We believe the forgetting problem is agnostic to the model architecture, and we went with 2-layer CNN as it was used by lots of previous work and to make experimentation easy.
> 4. This is a great suggestion for another experiment, in our experiment we pick a standard fixed client fraction as most of the previous works.
> 5. This is false as we experiment with FEMNIST that has 62 classes.
>
> `Federated Continual Learning (FCL) also focuses on the catastrophic forgetting problem. What’s the difference between the problem of this paper and that of FCL?`
>
> FCL has completely **different** problem formulation. We show that in standard FL formulation there is a forgetting problem. In FCL the Continual Learning problem formulation is adopted in FL where data changes over time (like CL).

---

### Note · Authors · 2024-12-11

I have read and agree with the venue's withdrawal policy on behalf of myself and my co-authors.